# Transalbugineal Artificial Urinary Sphincter: A Refined Implantation Technique to Improve Surgical Outcomes

**DOI:** 10.3390/jcm12083021

**Published:** 2023-04-21

**Authors:** Emilio Sacco, Filippo Marino, Carlo Gandi, Riccardo Bientinesi, Angelo Totaro, Stefano Moretto, Filippo Gavi, Marco Campetella, Marco Racioppi

**Affiliations:** Department of Urology, Fondazione Policlinico Universitario A. Gemelli IRCCS, Università Cattolica del Sacro Cuore, 00168 Rome, Italy

**Keywords:** urinary incontinence, male incontinence, artificial urinary sphincter, post-prostatectomy incontinence

## Abstract

The artificial urinary sphincter (AUS) implantation is an effective treatment of post-prostatectomy urinary incontinence (PPI). Still, it may result in troublesome complications such as intraoperative urethral lesion and postoperative erosion. Based on the multilayered structure of the tunica albuginea of the corpora cavernosa, we evaluated an alternative transalbugineal surgical technique of AUS cuff placement with the aim to decrease perioperative morbidity while preserving the integrity of the corpora cavernosa. A retrospective study was conducted in a tertiary referral center from September 2012 to October 2021, including 47 consecutive patients undergoing AUS (AMS800^®^) transalbugineal implantation. At a median (IQR) follow-up of 60 (24–84) months, no intraoperative urethral injury and only one noniatrogenic erosion occurred. The actuarial 12 mo and 5 yr overall erosion-free rates were 95.74% (95% CI: 84.04–98.92) and 91.76% (95% CI: 75.23–97.43), respectively. In preoperatively potent patients, the IIEF-5 score remained unchanged. The social continence (0–1 pads per day) rate was 82.98% (CI 95%: 68.83–91.10) at 12 mos and 76.81% (CI 95%: 60.56–87.04) at 5 yrs follow-up. Our technically refined approach to AUS implantation may help to avoid intraoperative urethral lesions and lower the risk of subsequent erosion without compromising sexual function in potent patients. Prospective and adequately powered studies are necessary to achieve more compelling evidence.

## 1. Introduction

Urinary incontinence is a common complication of prostatic surgery for benign and malignant prostatic disease [1], affecting 5–40% of patients after radical prostatectomy [2]. Artificial urinary sphincter (AUS) and perineal slings are the most used surgical treatment options for patients with persistent post-prostatectomy urinary incontinence (PPI) that fails conservative treatment [3,4]. European guidelines recommend the AUS implantation for men with moderate-to-severe PPI after unsuccessful conservative treatment. Conversely, for mild-to-moderate PPI, fixed perineal slings are recommended [5]. As a result, moderate urinary incontinence is in a ‘grey area’ in which both AUS and slings are indicated, even though there is growing evidence that AUS significantly outperforms fixed slings in men with moderate PPI [6].

The AUS was first introduced by Scott in 1973 [7]. The original AMS 721 model had many revisions and improvements, resulting in the current model AMS 800^®^ (Boston Scientific, Boston, MA, USA), which currently is the device with the longest follow-up and the largest amount of supporting literature [8]. Multiple case series with long-term follow-up have demonstrated a high success rate and high patient satisfaction with AUS implantation [9]. Nevertheless, AUS placement is an invasive procedure that can result in adverse events, such as intraoperative urethral lesions, postoperative infections, postoperative urethral erosions, and mechanical failure, that may require surgical revision or the explantation of the device [10,11]. In particular, intraoperative urethral injury is a very dreaded complication leading to early erosion and/or infection if unrecognized, or to procedure abortion with persistent incontinence (a redo surgery is usually performed after six months) and an increased risk of erosion after subsequent AUS implantation. 

The most common AUS cuff implantation site is the bulbar urethra at the level of the bifurcation of the corporal bodies [12]. A distal single or double cuff placement is often required in patients undergoing AUS reimplantation after urethral erosion or in those with urethral atrophy at the original cuff site [13]. Transcorporal AUS implantation has been described as a salvage surgical procedure to decrease the risk of urethral lesions and erosions in patients with damaged or frail urethra [14], or those undergoing first-time AUS placement after previous radiation therapy [15]. The major drawback of the transcorporal approach limiting its common use is the risk of impaired erectile function due to the interruption of the tunica albuginea of the corporal bodies [16]. 

Several histologic studies have described the tunica albuginea of the corpora cavernosa as a bilayered structure with multiple sublayers [17]. A histologic study showed the different histoanatomical patterns of the tunica albuginea. The most common architecture pattern consists of two layers: inner circular and outer longitudinal. More rarely, the tunica albuginea consists of three layers: inner circular, longitudinal, and outer circular [18]. Based on this anatomical evidence, we adopted a transalbugineal approach for cuff placement with the aim of avoiding intraoperative urethral lesions and possibly reducing the occurrence of postoperative urethral erosions (at least on the dorsal site of the urethra) without jeopardizing the integrity of the corpora cavernosa and the erectile function in potent patients. In this study, we described our technically refined approach to evaluating multiple outcomes, with a special focus on complications.

## 2. Materials and Methods

### 2.1. Patient Recruitment and Data Items

The study was conducted in our Italian tertiary referral center for male urinary incontinence management and received approval from the local Institutional Review Board (ID: 5094/2022). We queried our prospectively maintained database including consecutive patients undergoing AUS implantation (AMS 800^®^, Boston Scientific, Boston, MA, USA) from September 2012 to October 2021. Patients were selected for AUS implantation if suffering from moderate-to-severe (>2 pad use or >300 mL of urine loss daily), stable (>12 months), stress-prevalent incontinence and having a normal cognitive function and manual dexterity. We started to adopt the described technique in all cases not undergoing a transcorporal approach after the first 11 cases (and one intraoperative urethral injury). The study inclusion criteria were transalbugineal implantation and informed consent signing. The exclusion criteria were previous AUS, previous urethral lesions or erosions, implantation technique other than transalbugineal, and less than 12-months follow-up. Data collection was carried out by two independent researchers (F.M. and S.M.) not involved in the surgical procedures. The following data were collected: -preoperative data: age, body mass index (BMI), Charlson Comorbidity Index (CCI), history of diabetes mellitus, use of anticoagulants, previous irradiation and prostate or pelvic surgery, previous anti-incontinence procedures or urethrotomy, incontinence severity and quality-of-life impact, main findings of the preoperative work-out, and erectile function;-perioperative data: operative time, cuff size and location, type and location of the pressure-regulating balloon (PRB), and intraoperative complications;-postoperative data: catheterization time, pain score, the time interval from implantation to activation, postoperative complications, continence, and sexual outcomes.

### 2.2. Preoperative Evaluation

A baseline evaluation was performed, including medical history and physical examination. Cognitive function and manual dexterity were evaluated to identify patients not eligible for AUS placement. The preoperative workout included urethrocystoscopy to rule out urethral/bladder neck strictures and bladder abnormalities, and urodynamic investigation in patients with storage lower urinary tract symptoms (LUTS) or symptoms of bladder outlet obstruction. Incontinence severity was assessed quantitatively by 24 h pad use and 24 h pad weighing tests and subjectively by the International Consultation on Incontinence Questionnaire-Urinary Incontinence Short Form (ICIQ-UI SF) [19]. Patients using condoms were invited to use pads during the pad weighing test. ICIQ-UI QoL question and the EuroQol Group Questionnaire (EQ-5D-5L) [20] were used to evaluate the quality-of-life impact. Erectile function was assessed using the International Index of Erectile Function Questionnaire (IIEF-5) [21]. 

### 2.3. Patient Preparation and Surgical Technique

Patient preparation was performed following an internal protocol. Antiseptic washing was performed the day before surgery and trichotomy just before surgery. Patients were placed in a high lithotomy position, and chlorhexidine gluconate abdominal and perineal scrub (10 min) was performed. One-shot antibiotic prophylaxis (cephazolin and gentamicin) was administered. 

An abdominal–perineal dual-surgical approach was used [22]. All procedures were performed by the same senior surgeon (E.S.) under general anesthesia. The operative field and PRB were rinsed with antibiotic solutions. A 12 Fr transurethral catheter was inserted before the surgery. In cases of bladder neck contracture, patients were preferentially treated by adopting a two-stage approach with an endoscopic incision performed before the AUS implantation. A midline perineal incision was performed with a subsequent dissection until the identification of the bulbospongiosus muscle, which was divided in the middle. The corpus spongiosum was then gently exposed in its anterolateral aspect. Buck’s fascia was incised bilaterally, and the posterior dissection was performed starting 5 mm off the bulbar urethra on both sides (Figure 1 and Figure 2), entering the multilayered structure of the tunica albuginea of the corpora cavernosa, thus leaving its external layer attached to the posterior aspect of the urethra (Figure 3). A circumferential dissection of 2 cm in length was thoroughly completed. Entering the corpora cavernosa may sometimes occur during dissection; in this case, absorbable sutures were used to close the usually small infringements of the tunica albuginea. After assessing the urethral circumference with a measuring tape, the most appropriate cuff size was placed. Figure 4 and Figure 5 display a transversal view of transalbugineal cuff placement. The pump and a 61–70 cmH_2_O PRB, filled with 22 mL of saline solution, were implanted in the scrotum and peritoneum cavity, respectively, through the abdominal incision. At the end of the procedure, the system was cycled twice, and the pump was deactivated. Skin incisions were approximated with absorbable running intradermal sutures. A double cuff was implanted as the primary surgery in some patients with previous irradiation or as a salvage procedure in those with recurrent incontinence; the same transalbugineal technique was used for both cuffs’ implantation.

### 2.4. Postoperative Management and Follow-Up

The bladder catheter was removed on postoperative day two in all patients except those who underwent endoscopic urethral incision as an associated procedure, whose catheter was removed on day seven. Patients were discharged on day three, and the urinary sphincter was activated six weeks later. Subsequently, patients were evaluated at 3 mos, 12 mos, and yearly follow-up visits. 

### 2.5. Outcome Measures

Outcomes were evaluated by filling in the questionnaires during follow-up visits. The primary outcome was the rate of patients not experiencing intraoperative urethral injuries or spontaneous (noniatrogenic) postoperative erosions, evaluated at 12 mos and 5 yrs follow-up. Secondary outcomes were the rate of intraoperative, early (within two months) and late postoperative complications, pain (assessed on a 0–10 visual analog scale), overall erosion-free and reoperation-free (for any reason) rates, functional outcomes, and quality-of-life scores. A successful outcome was defined as achieving a cure (no pad use) or social continence (use of no more than one pad per day) [23]. Otherwise, patients were considered failures. A cure or social continence after device revision or replacement for recurrent incontinence were considered successful outcomes (although included the amount of complications), while AUS explantations because of erosion/infection were considered failures. Postoperative complications were classified based on the Clavien–Dindo classification [24]. 

### 2.6. Statistical Analysis and Reporting

Demographic, perioperative, and follow-up data were analyzed using descriptive statistic techniques. Quantitative variables were expressed as the median and interquartile range (IQR) or, otherwise, as mean ± standard deviation. Qualitative variables were reported as absolute and relative frequencies (percentages). The Wilcoxon test was used to compare differences in variables from the baseline. Kaplan–Meier survival analysis was used to assess both primary and secondary functional outcomes. Statistical significance was defined as two-sided *p*-values < 0.05. Statistical analyses were performed using version 14 of the STATA software (Stata Corp, College Station, TX, USA).

The study was reported in compliance with the Strengthening the Reporting of Observational Studies in Epidemiology (STROBE) guidelines (Appendix A) [25].

## 3. Results

### 3.1. Baseline Patient Characteristics

Figure 6 displays the study flowchart. Overall, 47 patients undergoing transalbugineal AUS implantation were analyzed. Table 1 shows the relevant baseline patient characteristics. Radical prostatectomy, open simple prostatectomy, and transurethral resection of the prostate were performed in 44 (93.6%), 1 (2.1%), and 2 (4.2%) patients, respectively. All patients were suffering from moderate-to-severe urinary incontinence (>2 pads/day) and failed previous conservative anti-incontinence treatments. Twenty-one patients (44.6%) had previous invasive treatment for stress incontinence (fixed sling or bulking agent). All patients had preoperative urethrocystoscopy, and 37 (78.7%) underwent urodynamic investigation as well.

### 3.2. Intraoperative and Early Postoperative Outcomes

Perioperative outcomes are summarized in Table 2. No intraoperative urethral lesions or other complications occurred. In eight patients that had a previously fixed sling implantation, none of the slings were removed. No patients had wound issues. Two patients presented scrotal hematoma that resolved spontaneously in two weeks. At postoperative day one, the median score of pain was two (0–10). Device activation was performed at 6 weeks. Overall, three (6.38%) patients had Clavien–Dindo grade ≥ 2 early complications: two cases of migrated pumps necessitating surgical repositioning and one case of liquid leakage from the connector between the pump and PRB.

### 3.3. Late Complications and Primary Outcome

At a median (IQR) follow-up of 60 (24–84) months, urethral erosion occurred in three patients who underwent device explantation and concomitant urethroplasty. An 82-year-old man suffering from post-radical prostatectomy incontinence, hypertension, and chronic heart ischemic disease, without previous irradiation, experienced a noniatrogenic urethral erosion at 24 mos follow-up. Another patient had urethral erosion five months after implantation due to catheterization in the emergency department because of a stroke episode. The third urethral erosion occurred in a 76-year-old man five years after AUS implantation and four weeks after a second radiotherapy treatment for local tumor recurrence. Overall, six patients (12.7%) presented with Clavien–Dindo grade ≥ 2 late postoperative complications: specifically, three urethral erosions, one patient underwent cuff relocation eight years after AUS implantation, and two patients with initial failure that were surgically rescued with the addition of a second cuff seven and four years after AUS implantation, respectively. Consequently, only one event occurred according to our primary outcome definition, with a 12 mo and 5 yr event-free survival rate of 100% and 97.2% (95% CI: 81.87–99.61), respectively (Figure 7). All erosions were treated with device explantation, which was performed only in these cases. As a result, the overall erosion-free survival rates matched the AUS actuarial survival rate (Figure 8): 95.74% (95% CI: 84.04–98.92) and 91.76% (95% CI: 75.23–97.43) at 12 mos and 5 yrs, respectively. The overall reoperation-free (for any reason) survival rate was 89.58% (95% CI: 76.77–95.53) at 12 mos and 82.51% (95% CI: 65.80–91.55) at 5 yrs follow-up (Figure 9).

### 3.4. Functional Outcomes

Functional outcomes are summarized in Table 3. A statistically significant improvement over the baseline was observed for all functional and quality-of-life outcomes at 12 mos follow-up, which were substantially confirmed at 5 yrs. During follow-up, two patients (4.25%) reporting storage LUTS were successfully treated with botulinum toxin-A intradetrusor injections after the failure of antimuscarinic therapy.

The actuarial cure rate was 46.81% (95% CI: 32.17–60.16) at 12 mos and 44.21% (95% CI: 29.67–57.79) at 5 yrs follow-up (Appendix A).

The social continence rate was 82.98% (95% CI: 68.83–91.10) at 12 mos and 76.81% (95% CI: 60.56–87.04) at 5 yrs follow-up (Appendix A). Most (65%) of the socially continent patients used a pad per day for security purposes because of occasional dribbling.

Sexual function was evaluated in 12 patients that were preoperatively potent. All patients remained potent postoperatively, and no worsening of their erectile function was observed. No statistically significant deterioration over the baseline was observed for the IIEF-5 score at 12 mos and 5 yrs follow-up.

## 4. Discussion

The AUS remains the gold standard treatment for male urinary stress incontinence, offering a very satisfactory and predictable continence rate and high patient satisfaction [27]. However, AUS implantation is associated with a likely underreported risk of intraoperative urethral lesions and postoperative urethral erosions [28]. To the best of our knowledge, this is the first study describing transalbugineal AUS placement to decrease the occurrence of urethral complications. In our series, three urethral erosions occurred, with only one noniatrogenic erosion and no infection or intraoperative urethral lesions.

The data about infection and erosion rates after AUS placement are inconsistent, with most papers not reporting the erosion and infection rates separately, counting them together as a composite outcome. In a pooled analysis [28] of 12 studies, including 562 patients, the mean infection plus erosion rate was 8.5% (3.3–27.8%). Our overall erosion rate (6.38%) is close to the lower limit of the literature without the occurrence of prosthetic infection. Excluding two patients with a clear inciting cause of urethral erosion (catheterization and redo pelvic irradiation), only one patient experienced a noniatrogenic urethral erosion. According to Cheung et al. [29] and Ortiz et al. [30], we think it is relevant to distinguish between the iatrogenic and noniatrogenic nature of the erosions because the former is reasonably poorly related to the surgical technique and may mask the possible advantage coming from the adoption of a novel implantation modality aiming to decrease the erosion/infection rate. Our rate of noniatrogenic erosions (2.1%) compares favorably with that reported by Ortiz et al. [30] in both a conventional technique series (6.1%) and a transcorporal implantation series (18.3%), and is close to that reported by Cheung et al. [29] using a dorsolateral tissue-preserving approach (0.9%).

Most intraoperative urethral injuries occur at a 12-o’clock position; that is the most difficult site of dissection because the urethra is thinner and adherent to the corpora cavernosa at this site. An unrecognized intraoperative urethral lesion is a very harmful event that results in early cuff erosion and infection. In case of intraperitoneal placement of the PRB, peritonitis may also occur. Due to unrecognized urethral injuries and reporting biases, this complication is likely largely underreported. The need to dissociate the urethra of the corpora cavernosa, especially in case of fragile and fibrotic urethra (e.g., prior irradiation or incontinence surgery), may weaken the posterior urethral wall and also potentially lead to further urethral devascularization, predisposing to intraoperative urethral lesions and future erosions. Thus, we hypothesized that leaving an extra layer of tunica albuginea of the corpora cavernosa attached to the posterior bulbar urethral wall may be beneficial, also increasing the surgeon’s awareness during the procedure. Furthermore, we implemented a standardized protocol of asepsis to avoid infection, including dual single-shot antibiotic prophylaxis, minimum air exposure time of the device components, limitation of the operating room traffic, use of double gloves by the members of the surgical team with frequent gloves changes, as well as rinsing of the perineal field and device components with antibiotic solution, use of medicated drapes to isolate the skin, meticulous hemostasis, and minimization of tissue dissection. 

All our erosions occurred in the ventrolateral position, in compliance with a previous report also evaluating the transcorporal approach [30]. Even though the periurethral tissue-preserving approaches should provide a bolstering effect on the dorsal urethra to protect against intraoperative injury and postoperative erosion, they did not protect the ventrolateral urethra. Albeit the ventral area is the thickest part of the urethra, it is also the most exposed to the external perineal pressure and damage from traumatic catheterization.

The overall reintervention rate of 19.1% also compares favorably with the literature: in a pooled analysis of 549 patients (10 studies) [28], the mean reintervention rate was 26% (range 14.8–44.8%). 

Over the past decade, different surgical approaches have been described for AUS implantation. Transcorporal AUS implantation has been described as a salvage surgical technique in patients with a damaged or frail urethra, with the aim to mitigate the risk of urethral erosion [13]. The transcorporal technique specifically interposes ventral corporal cavernosa tunica albuginea between the cuff and the urethra and thereby theoretically reduces the risk of erosion [31]. Miller et al. reported that transcorporal AUS placement resulted in a significantly lower number of major complications, explants, and revisions in patients with a history of prior pelvic radiation [15]. Conversely, in a large, multi-institutional, prospective study, Brant et al. demonstrated that men who underwent transcorporal AUS implantation had similar rates of explantations than those not receiving this surgical approach. Although the transcorporal placement was not statistically protective from eventual erosion, the overall explantation rate in the high-risk population was lower than in patients not receiving the transcorporal approach [32]. The most commonly cited pitfall of transcorporal AUS implantation is a violation of the tunica albuginea of the corporal bodies, which can result in erectile dysfunction [14,15,33]. 

The data regarding sexual function in AUS patients are scarce in the literature, probably because of the high prevalence of erectile dysfunction secondary to prostatic surgery or radiation therapy. Few studies on transcorporal AUS cuff placement analyzed erectile function outcomes. Theoretically, the incision of the corpora cavernosa performed during the transcorporal approach may cause loss of erectile rigidity, although a few authors consider the transcorporal AUS implantation as a sexually safe procedure. Indeed, Brant and Martins argue that closure of the corporal body may prevent postoperative bleeding as well as loss of erectile rigidity [34]. However, a long-term analysis of 39 patients undergoing transcorporal AUS implantation showed that erectile function was present in 15.4% of the patients preoperatively, compared to only 7.7% after transcorporal AUS cuff placement [35]. While a low rate of intraoperative urethral lesions and noniatrogenic urethral erosions was observed in our series, even in a population with a high rate of previous irradiation or urethrotomy, none of the preoperatively potent patients experienced a deterioration of erectile function. Of note, the transcorporal technique is not suitable for all patients but is a useful alternative to salvage surgery in patients with a very fragile urethra. Lee et al., in their surgical practice, offered transcorporal AUS cuff placement to patients who were impotent preoperatively and who were not interested in the resumption of erectile function [36]. Furthermore, while the transcorporal approach may limit patient eligibility to undergo penile prosthesis implantation [30], our modified technique does not prevent the patient from performing this surgery. Vice versa, unlike the transcorporal technique, the transalbugineal approach is theoretically also possible in patients with penile prosthesis, although extreme caution is required. 

Serra et al. [37], in 2016, described a bulbospongiosus-muscle-sparing AUS implantation technique, with the idea that muscle preservation may decrease the risk of urethral injury during the dissection and may also better preserve the flow of blood to the urethra, thus preventing both long- and short-term complications. In this series of 82 consecutive patients, Serra et al. reported an AUS survival rate, defined as the loss of continence or mechanical failure during follow-up, of 95.5% (95%CI: 89.4–100%) at 24 mos and 62.6% (95%CI: 45.5–79.6%) at 60 mos, similar to our findings, while somewhat better functional outcomes were reported (dry rate of 76.8% and social continence rate of 92%). No intraoperative complications occurred, and four patients required re-operation because of urethral erosion, device infection, and pump and cuff relocation.

Our refined implantation technique demonstrated to be an effective, safe, and reproducible procedure, with continence rates similar to those achieved when using a conventional approach [28,38,39,40]. Definitions of continence based on pad use are heterogenous in the literature; however, in AUS publications, “social continence”, meant as the use of at maximum one pad per day, is the most used quantitative functional outcome. In our series, 78.7% of patients were socially continent at 5 yrs follow-up, representing a good functional result compared with the existing literature. The critical systematic review by Van der Aa et al. [28] reported a social continence rate of 79.0% (60.9–100%), based on data from seven studies, including 262 patients. Furthermore, our results need to be interpreted accounting for the quite high median age of our cohort; indeed, studies have been published reporting that AUS implantation in elderly men could be associated with poorer clinical outcomes [41].

Urethral atrophy is a late complication, usually presumed when incontinence recurrence occurs during follow-up with a functioning sphincter [40]. The commonly presumed pathophysiology underlying urethral atrophy is hypoxia of urethral tissue, which is attributed to the long-standing pressure exerted by the cuff on the urethral wall [42]. Other authors reported a mean urethral atrophy rate of 7.9% (1.9–28.6%) at a median time of 30 months [28]. With our technique, we observed only two patients (4.25%) who developed urethral atrophy, requiring surgical revision with a second cuff placement; however, a longer follow-up and larger sample size are required to draw a more convincing conclusion on this outcome. The occurrence of atrophy (and erosion) may be related to the cuff size; based on the cuff sizes used in our series, our modified approach did not translate into the implantation of larger cuffs compared to the previous series using the usual implantation technique, in which 4.0 and 4.5 cm were the most used cuff sizes [39,43].

Our study has several strengths: the prospectively maintained database of consecutive patients, single-surgeon experience using a standardized technique, long-term follow-up, and use of multiple outcomes including pad weight that are recognized as the most accurate metrics for UI assessment [44]. Nevertheless, this study is not devoid of limitations as the retrospective design, the single-center nature, and the limited sample size preclude the authors from making definitive recommendations. Given the low incidence of male stress incontinence and the rarity of the study event (erosion), several hundreds of patients are necessary to demonstrate statistically the superiority in outcomes of a novel AUS implantation technique. We aimed to describe a technique that should improve the safety of the urethral dissection and that achieved long-term satisfactory results even in a high-risk population.

## 5. Conclusions

Our study is the first to describe the AUS transalbugineal placement, showing that it is a technically feasible, safe, and effective procedure. Our modified approach may lower the occurrence of urethral erosions and prosthetic infection by avoiding shallow urethral dissection and increasing surgeon awareness of often unrecognized urethral injuries. However, prospective and adequately powered, possibly multicenter, studies are necessary to achieve more compelling evidence.

## Figures and Tables

**Figure 1 jcm-12-03021-f001:**
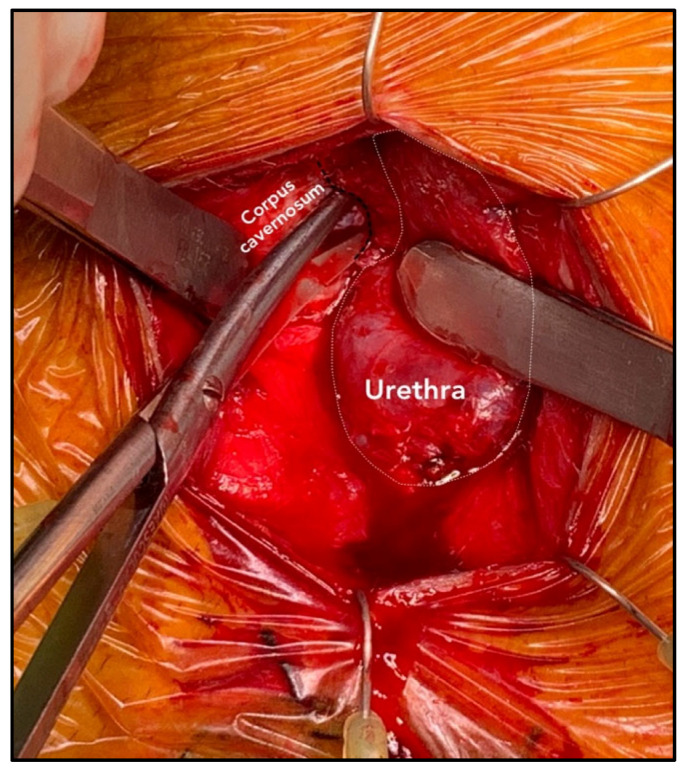
Left incision of the tunica albuginea of the corpora cavernosa 5 mm off the posterior attachment of the urethra to the corpora cavernosa.

**Figure 2 jcm-12-03021-f002:**
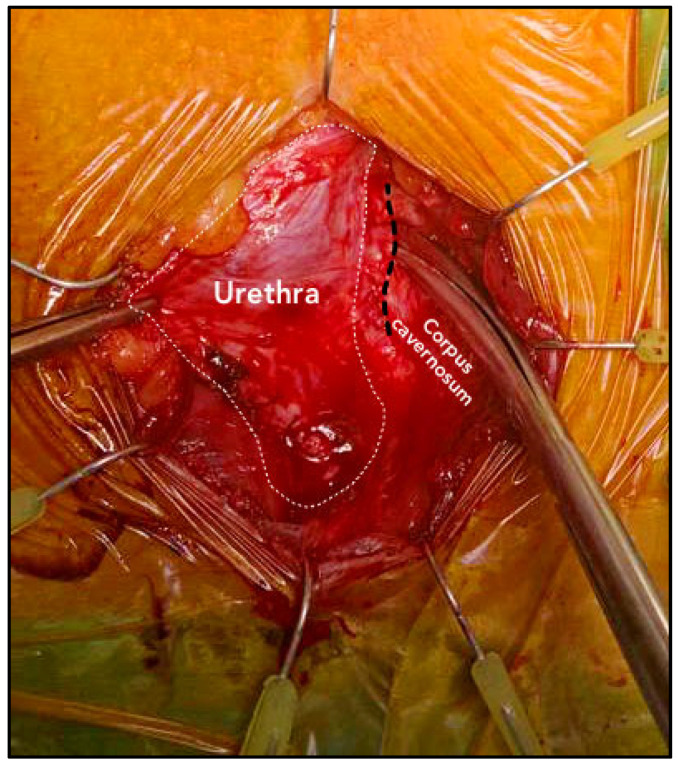
Right incision of the tunica albuginea of the corpora cavernosa 5 mm off the posterior attachment of the urethra to the corpora cavernosa.

**Figure 3 jcm-12-03021-f003:**
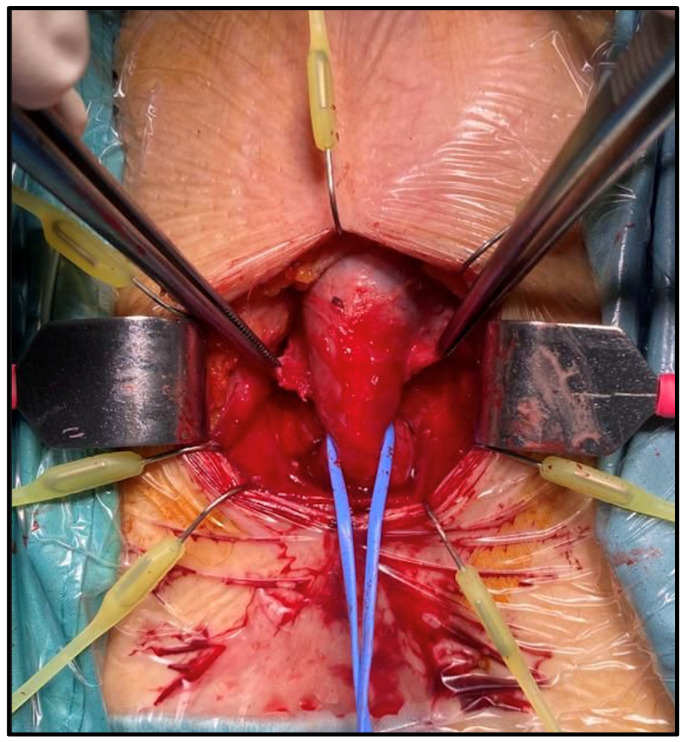
Urethra after completed dissection with attached external layer of tunica albuginea.

**Figure 4 jcm-12-03021-f004:**
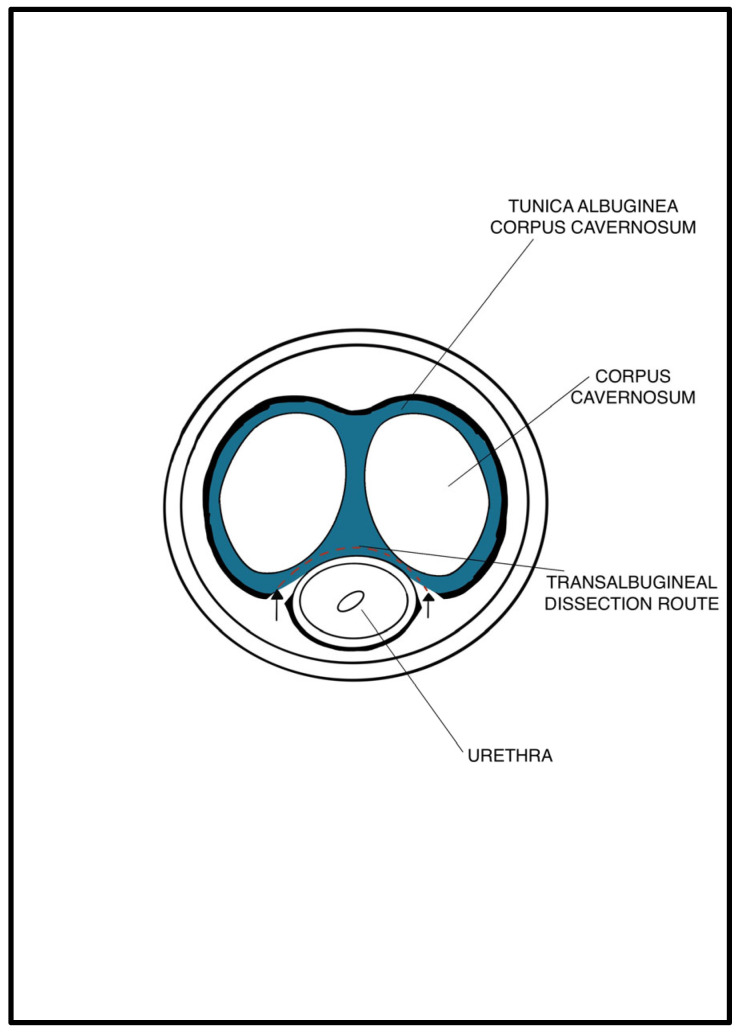
Transversal view of transalbugineal dissection.

**Figure 5 jcm-12-03021-f005:**
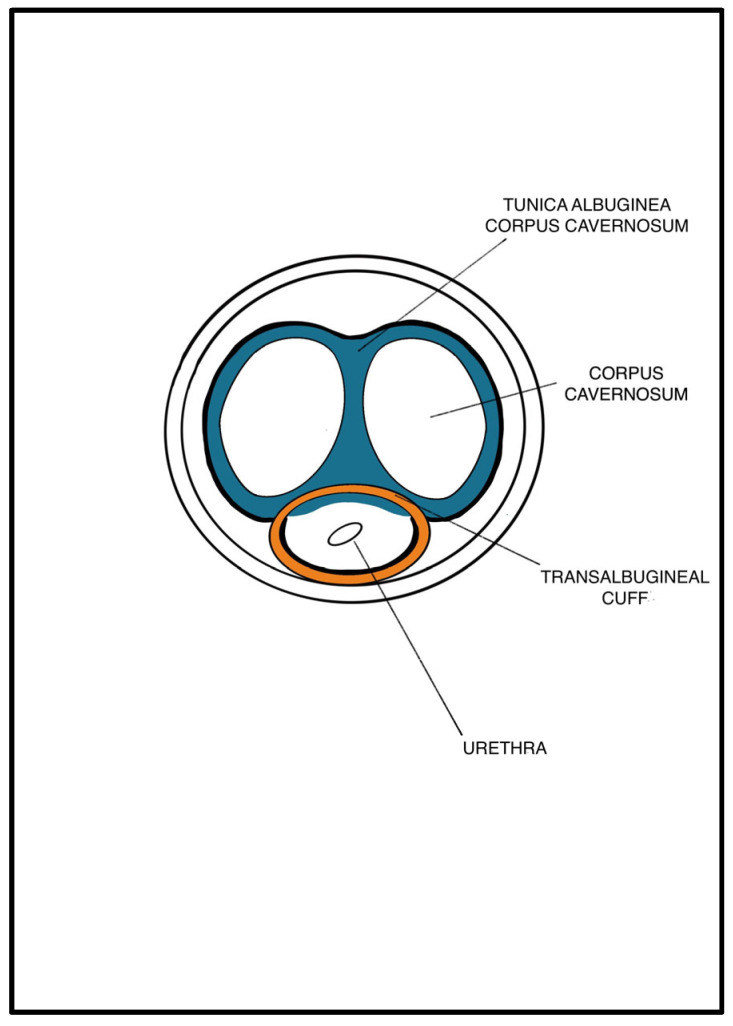
Transversal view of transalbugineal cuff in site.

**Figure 6 jcm-12-03021-f006:**
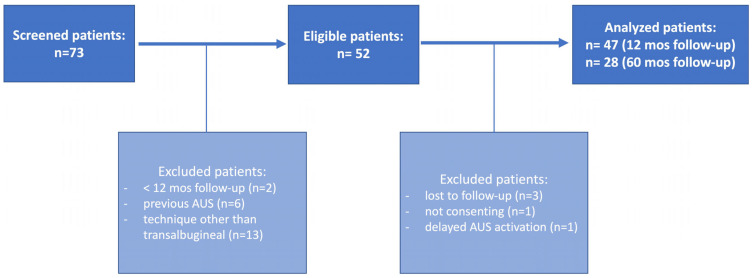
Study flowchart.

**Figure 7 jcm-12-03021-f007:**
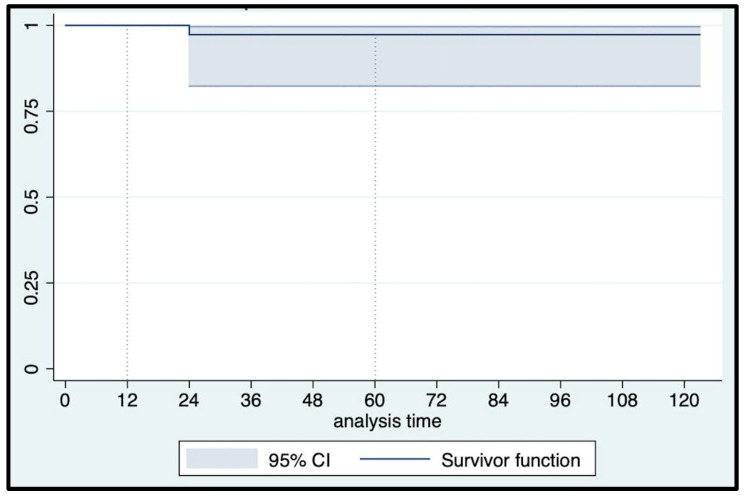
Kaplan–Meier noniatrogenic erosion-free survival curve.

**Figure 8 jcm-12-03021-f008:**
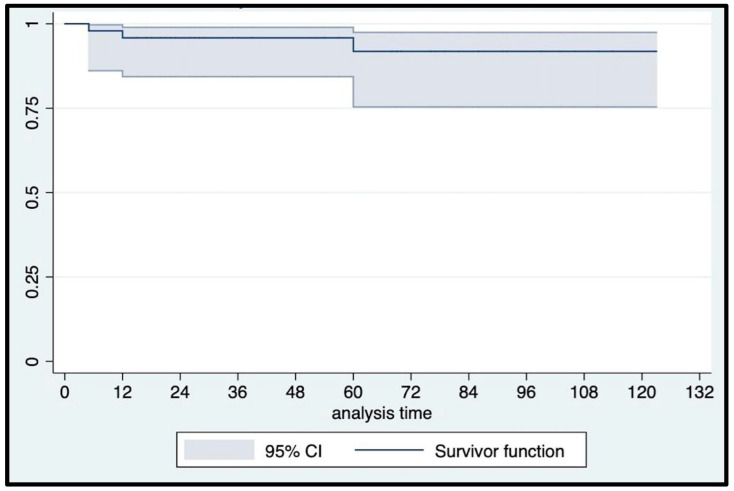
Kaplan–Meier overall erosion-free survival curve.

**Figure 9 jcm-12-03021-f009:**
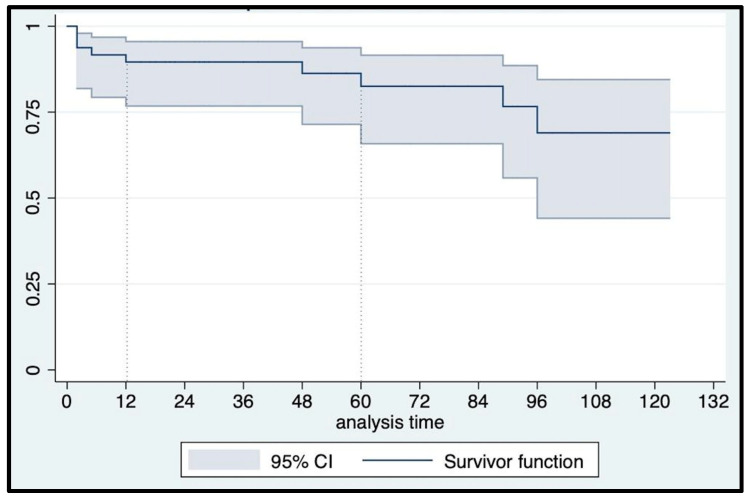
Kaplan–Meier reoperation-free survival curve.

**Table 1 jcm-12-03021-t001:** Patient baseline characteristics (n = 47): (data are reported as the median and IQR or number and percentage).

Baseline Characteristics	
Age, years	76 (72–78)
BMI, kg/m^2^	28 (25–30)
CCI	3 (2–3)
Diabetes mellitus, *n* (%)	3 (6.38)
Oral anticoagulant/antiplatelet, *n* (%)	18 (38.29)
Median duration of UI, mos	60 (36–84)
Median 24 h PAD weighing test, grams	750 (655–1200)
Median 24 h PAD use, *n*	6 (5–6)
UI clinical type, *n* (%)	
SUI	39 (82.97)
MUI	8 (17.02)
Previous prostate surgery, *n* (%)	
RARP	12 (23.53)
RRP	24 (51.06)
LRP	8 (17.02)
TURP	2 (4.25)
OSP	1 (2.13)
Previous UI procedures, *n* (%)	
Fixed sling ^1^	8 (17.02)
Bulking agent	13 (27.65)
Botulinum toxin	2 (4.25)
Previous pelvic RT, *n* (%)	22 (46.80)
Previous urethrotomy, *n* (%)	18 (38.29)
Main urodynamic findings, *n* (%) ^2^	
Detrusor overactivity	9 (24.32)
Detrusor underactivity	13 (35.13)
Bladder outlet obstruction	1 (2.70)
Stress incontinence	37 (100)
Maximum cystometric capacity, cmH_2_O	300 (180–400)

^1^ Most (6) of the slings were TILOOP [26]. ^2^ Thirty-seven patients underwent urodynamic evaluation. BMI, body mass index; CCI, Charlson Comorbidity Index; UI, urinary incontinence; SUI, stress urinary incontinence; MUI, mixed urinary incontinence; OSP, open simple prostatectomy; RARP, robot-assisted radical prostatectomy; RRP, retropubic radical prostatectomy; LRP, laparoscopic radical prostatectomy; TURP, transurethral resection of the prostate; RT, radiotherapy.

**Table 2 jcm-12-03021-t002:** Intraoperative and early complications data.

Intraoperative Data	
Median operative time (IQR), min	90 (70–120)
Cuff size, *n* (%)	
3.5 cm	2 (4.25)
4.0 cm	14 (29.78)
4.5 cm	24 (51.06)
5.0 cm	7 (14.89)
Single cuff, *n* (%)	43 (91.48)
Double cuff, *n* (%)	4 (8.51)
Associated endoscopic uretrothomy, *n* (%)	3 (6.38)
Early complications	
Scrotal hematoma, *n* (%)	2 (4.25)
Migrated pump, *n* (%)	2 (4.25)
Liquid leakage from the connector, *n* (%)	1 (2.12)
Clavien–Dindo early complications grade, *n* (%)	
Grade 1	14 (29.79)
Grade ≥ 2	3 (6.38)

**Table 3 jcm-12-03021-t003:** Functional outcomes (numbers indicate mean values with standard deviation and *p*-values of comparison with the baseline).

Time Point	24 h Pad Number	24 h Pad Weight	ICIQ-SF	ICIQ-QoL	EQ-5D-5L	IIEF-5 *
Baseline (n = 47)	5.85 ± 1.99	984 ± 557.35	17.97 ± 2.21	8.4 ± 1.48	72 ± 14.9	13.25± 3.83
12 mos (n = 47)	0.72 ± 0.77 (*p* < 0.0001)	21.48 ± 40.68 (*p* < 0.0001)	4.60 ± 4.61 (*p* < 0.0001)	1.78 ± 2.47 (*p* < 0.0001)	81 ± 14.2 (*p* = 0.0002)	13.75 ± 3.93 (*p* = 0.76)
5 yrs (n = 28)	0.84 ± 0.89 (*p* < 0.0001)	25.46 ± 46.03 (*p* < 0.0001)	4.46 ± 4.83 (*p* < 0.0001)	2.28 ± 2.85 (*p* < 0.0001)	76 ± 12 (*p* = 0.0002)	13.75 ± 3.93 (*p* = 0.76)

* Evaluated in only 12 patients that were preoperatively potent. ICIQ-SF, International Consultation on Incontinence Questionnaire-Urinary Incontinence Short Form; ICIQ-QoL, International Consultation on Incontinence Questionnaire- Quality of Life question; EQ-5D-5L, EuroQol Group Questionnaire; IIEF-5, International Index of Erectile Function Questionnaire.

## Data Availability

Data related to patients and surgical procedures can be found at the Department of Urology, Fondazione Policlinico Universitario A. Gemelli IRCCS–Università Cattolica del Sacro Cuore, Largo Agostino Gemelli 8, 00168 Rome, Italy.

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
