# Peer review of "Transalbugineal Artificial Urinary Sphincter: A Refined Implantation Technique to Improve Surgical Outcomes"

_jcm, 2023, doi:10.3390/jcm12083021_

Round 1

Reviewer 1 Report

This is a retrospective chart review of a novel technique by a single surgeon in management of stress urinary incontinence with AUS utilizing a transalbugineal approach to placement of the cuff with the goal of evaluating whether this decreased morbidity and preserved erectile function. The authors set out to look at short- and medium-term outcomes of erosion, continence, and erectile function. Participants were required to have at least 12 months of follow-up for inclusion in analysis. The authors should be applauded for thinking of novel techniques to help decrease the risk of AUS erosion which is a major drawback of utilization of this device and a huge impact to patients. These are important data to present and vital to distribute this technique to a larger audience.

I think the biggest concern that I have is how the authors classify erosion – the authors define erosions as “spontaneous (no apparent external causes)”, but this is not necessarily the best way to classify erosions, and I believe erosions should be counted regardless of if there was an external cause. Often erosions occur when there is some inciting factor such as catheterization, and these are important to include, and have been included in previous literature. Not including these erosions within this dataset makes it very difficult (and a bit misleading) to compare these outcomes to the existing literature for this reason. Though there were 3 erosions, the authors only include 1 of those events in the 5-year survival rate and none in the 12-month survival rate, despite the fact that one of these erosions occurred at 5 month. Just because there were other factors (catheterization, patient receiving radiation) should not exclude them from this outcome and I think the analysis needs to be re-done, including all erosions regardless of whether there was an apparent “external cause”, as has been done previously in the vast majority of the literature.

In addition, I did not see a power calculation included so it is difficult to interpret the results without that knowledge.

A few other notes -

In my experience, the AUS cuff is typically not implanted “proximal to the bifurcation of the corporal bodies” – perhaps the authors meant distal to the bifurcation?

It would be helpful to understand if this technique resulted in use of larger cuff sizes than a standard approach. Perhaps one way to get at this would be to compare the average cuff size used to the surgeon’s previous cohort before implementation of this new technique. It would be very interesting to see that this results in larger cuff sizes.

There is a patient that did not have his device activated until 12 months because of other reasons – I believe this patient should be excluded from analysis given that he did not have activation at 6 weeks and thus not on the same trajectory as the other patients.

The authors note that erosions typically occur at 12 o’clock although I believe there is conflicting evidence around this. When erosions occurred within this cohort, did they occur ventrally or dorsally? It would be very interesting to compare to any existing literature (or pre-novel technique intervention) if possible. This could be added evidence of the significance of this novel technique specifically in addressing dorsal erosion, even if it may not change the incidence of ventral erosion. ***

This technique may be useful in patients with concomitant ED who want an IPP/penile prosthesis. Often if patients have concomitant ED/SUI and they have an erosion, there is concern about placing a transcorporal cuff because of risk of injury to an existing penile implant, or because it will make future penile implant challenging. This could be a very good application of this technique and could be highlighted in the discussion.

Small note – in the study flow-chart eligible is mis-spelled

Overall I commend the authors on the development of this new technique and feel that this is important data to share and will be of interest to urologists managing SUI. I believe there are important changes that need to be made regarding the classifications of erosion as noted above before this should be published.

Reviewer 2 Report

Dear authors

i congratulate to the efforts to evaluate a new implantation technique for the AMS 800. 

I have the following comments to the manuscript: 

Methods: 

Please describe more the minimum requirement (UI severity grading) for patients receiving and AUS. 

You descripe in the abstract that this was a retrospective study, in the methods that it is prospective. Please revise and add the correct investigation. 

Please explain why are you referring to “spontaneous” (no external cause) erosions, What do you define under these circumstances? And why do you exclude these? In large series, there are some cases with iatrogen injuries due to catheterization, however, the majority develop eereosiosn despite of being catheterized. Theese series always include all erosions, independently from the cause. In the result section you finally refer to a total of 3 erosions including also itraogenic cuases. 

Do all patients receive in your center transalbugineal implantation? If not, how do you select? Please explain in the methods section. 

Figure 6: Please point out more clearly: How many patients available for 12 months follow up, how many for 60 months. 

Reesults: 
The median age of is surprisingly high. Please comment in the discussion section. Could this be due to selection bias? 

Tabel 3: Please add the number of patients with complete results to evaluate  you have for each section

I suggest to remove the cure and continence failure free and event free survival curves. There are too many of the same curves, limited interpretation due to small sample size. 

Discussion: You are referring again to “only spontaneous” erosions. However, the literature you are referring to did not exclude iatrogen erosions, neither. Please critical discuss the actual outcome (not only spontaneous ones) with the literature. 

You are referring to intraoperative urethral injuries as to a major factor for postoperative erosions. I would expect these erosiosn to occur in a timely manner after surgery. However, the literature does not provide many evidence that there are many intraoperative urethral injuries during the surgery (particular not in primary implantation). Please explain how you support your hypothesis based on literature, since I did not spot the literature and according numberst. 

Furtheermore, I am missing the critical discussion, that the erosions rates do acutally not suffer substantially to the “classic” implantation way. This may be referred to the limited number of patients, however, I am struggeling of seeing the benefit of the procedure. Please revise and discuss more critically. 

The long-term survival is impressive, I would suggest to more critical discuss your long-term outcome and possible causes?! 

How does the increased age could bee influenced for this outcome? (Reduced activity due to advanced age (e.g. no bycycle, no running, no sauna etc. which may adversely affect the AUS?)

Please end the manuscript with a conclusion and not the limitations. 

I would suggest to soften the possible impact of the surgical technique, since there are many variables and limited patient numbers to provide any general recommendation. 

Round 2

Reviewer 2 Report

Dear authors,

all comments have been addressed. 

I have just a last comment on the abstract: There is still described "A retrospective study was conducted in a tertiary referral center from ...." 

I understood from your methods section revision, that this was an prospectively maintained database. The abstract would still benefit from this information instead of retrospective study,